# OpenReview forum: "SHARP: Structured Hierarchical Attention Rank Projection for Efficient Language Model Distillation"
_ICLR.cc/2026/Conference — ICLR 2026 Conference Withdrawn Submission_

### Official Review · Reviewer_v3bW · 2025-10-27

**Soundness:** 3
**Presentation:** 3
**Contribution:** 2
**Rating:** 2
**Confidence:** 4

**Summary:**

The authors propose a novel distillation method for transformers. They introduce attention patterns at different levels and use them for model distillation. To avoid interference among the losses corresponding to different attention pattern levels, the authors propose to use the method of orthogonal rank space projection to make the gradient directions as orthogonal as possible.

**Strengths:**

1. The paper has a strong point: it uses attention values for distillation learning, rather than being largely limited to only using the model’s output logits for distillation like traditional algorithms.
2. The paper conducts extensive experiments and verifies the effectiveness of the proposed method from multiple perspectives.

**Weaknesses:**

1. Some parts of the paper lack detailed descriptions, such as the role of the distillation loss for \(H_{\text{int}}\) and \(H_{\text{init},T}\); for specific details, refer to the "Questions".
2. There are inconsistencies in certain parts of the method section throughout the paper; for specific cases, refer to the "Questions".

**Questions:**

1. The "Gradient Interference" is minimized when the two gradients are in the same direction, rather than being orthogonal. However, the paper seems to take gradient orthogonality as the objective.

2. The "Gradient Interference" in Proposition 1 is inconsistent with that in Definition 1. "Gradient Interference" as defined in Definition 1 refers to the difference between gradients of different losses with respect to the **same parameter** $\theta$, rather than the difference between derivatives with respect to **different parameters** ($W^{g_i}$ and $W^{g_j}$) as shown in Proposition 1 (i.e., $\nabla_{W^{g_i}}\mathcal{L}^{g_i}$ and $\nabla_{W^{g_j}}\mathcal{L}^{g_j}$). What we are actually interested in is $\bigl\lvert \langle \nabla_{\theta}\mathcal{L}^{g_i}, \nabla_{\theta}\mathcal{L}^{g_i}\rangle \bigr\rvert$, where $\theta$ denotes the parameters of the student model itself.

3. For $\phi_g$ (i.e., pooling + projection) in Line 206, no explicit mathematical definition is provided. Specifically, it is unclear along which dimension(s) the pooling operation is performed. Additionally, how does this projection map the variable-length $A$ to a fixed dimension $D$? Specific descriptions of these implementation details is missing.

4. Equation 10 is inconsistent with Algorithm 2. Algorithm 2 includes an additional term $\beta \times L_{\text{int}}$, and $L_{\text{ortho}}$ is missing.

5. In the "Computational Overhead" paragraph of Section 4.4, the parameters of the MHA used to compute $H_{\text{int}}$ do not seem to be included in the overhead calculation. Based on my understanding, this MHA also contains additional trainable parameters that should be accounted for.

6. How is the effective capacity ratio in Section 4.6 calculated? Furthermore, $\rho$ in the corresponding formula is not defined.

7. According to the pipeline in Figure 2 and the subsequent introduction of MHAs, if both MHAs are trainable, and only $\bigl\| H_{\text{int}} - H_{\text{int,T}} \bigr\|^2$ is used as the loss for training these MHAs, then the two MHAs should directly converge to output identical values (thus minimizing this loss to 0).

---

### Official Review · Reviewer_ib5F · 2025-10-27

**Soundness:** 3
**Presentation:** 2
**Contribution:** 3
**Rating:** 6
**Confidence:** 4

**Summary:**

The paper introduces new method for knowledge distillation training named SHARP (Structured Hierarchical Attention Rank Projection). It extends classical knowledge distillation by transfering attention patterns of transformer models across token-, head-, and layer-level granularities to improve performance. Further, the authors employ orthogonal rank space projections to disentangle these components into independent subspaces. The method is evaluated on both language generation and understanding tasks across multiple model sizes for teacher and student models.

**Strengths:**

- The proposed method is fairly simple by explicitly using different parts of the attention mechanism in the knowledge distillation process. The related theorhetical discussion clarify on many aspects such runtime overhead or the impact on gradient interference from the orthogonal rank projections.
- The paper presents promising gains of more than 20% over recent baselines on DollyEval, but also clear improvements in NLU tasks compared to other knowledge distillation methods. The results for NLG might be better too, but some clarification is required here. See also questions below.
- The experiments show that SHARP works across different model sizes, provides training speed ups due to faster convergence, and also work for coding settings.

**Weaknesses:**

- Figure 1 is hard to understand from the limited information provided. What are token, span, paragraph, sentence based KD methods. This get clarified later in the paper but causes confusion in the first part. Further, after understanding what token, head and layer stands for, what do you want to show with the tSNE graph? The objective of SHARP targets at aiming to seperate these representations and we see that it works but I rather would like to see some qualitative example, e.g., what kind of attention pattern the model is now able to learn (or in other words, which it did not learn using classical KD).
- I am missing ablations study, particularly on questions why this approach works. One of the core takeaways for me is that we see that teaching the student softmax-attention is crucial, however, why are we going for token, head and layer attenion aggegration? Especially for token or head attention, experiments could show that SHARP does teach the student certain patterns that classical KD does not.
- I think the theorhetical part reads a little bit disconnected from the experiment section, e.g., one of the core claims is that SHARP avoids gradient interference, however, the empirical results on this claim are rather short or limited to a few sentences. The core empirical results are about improving downstream performance. While the results are good, the flow of the paper could be improved by either (1) aligning the theorhetical part with the empirical part or (2) add some qualitative insights (see above)

**Questions:**

- Can you please clarify what metrics you report for NLG? In the text you mention it is perplexity, so lower would be better, but in the table SHARP you highlight the highest (the best or worst?) scores.
- Some methods appear only in related work but in the baseline section, you might want to add MiniLLM, etc. here too.
- Do you have insights on leaving certain aggregation (token, head, layer) out of the optimization process and its impact on the downstream performance?

---

### Official Review · Reviewer_2teV · 2025-10-30

**Soundness:** 3
**Presentation:** 3
**Contribution:** 3
**Rating:** 6
**Confidence:** 3

**Summary:**

This paper SHARP (Structured Hierarchical Attention Rank Projection) , a knowledge distillation framework designed to compress large language models by transferring multi-granular attention knowledge (token, head, layer) from teacher to student models. The key innovation lies in projecting these attention patterns into orthogonal rank spaces, thereby eliminating gradient interference across granularities during optimization. Extensive experiments on NLG and NLU tasks show consistent improvements over baselines, especially when distilling from larger teachers (up to 6.7B) into a 125M student, achieving up to 7.2% perplexity reduction and 2–3× faster convergence.

**Strengths:**

1. The introduction of orthogonal rank space projections effectively decouples multi-granular distillation objectives, mitigating gradient interference—a long-standing issue in multi-objective optimization.
2. SHARP consistently outperforms strong baselines across diverse model families (OPT, LLaMA, DeepSeek) and tasks (NLG/NLU), with gains scaling favorably with teacher size.
3.  The paper provides formal analysis (Theorems 1–2) quantifying gradient interference reduction and capacity gains, lending credibility to the approach beyond empirical success.

**Weaknesses:**

1. The study restricts granularities to token/head/layer (|G|=3); it remains unclear how performance scales or degrades with more/fewer granularities or adaptive granularity discovery.
2. While convergence is faster, each epoch incurs 25–30% extra compute and ~0.9M extra parameters; cost-benefit analysis for very large models or edge deployment is not discussed.
3. Comparisons omit recent layer-wise or block-wise distillation methods that also address interference (e.g., DKD, LRC-KD), making the claimed superiority less definitive.

**Questions:**

N/A

---

### Official Review · Reviewer_mBc9 · 2025-10-30

**Soundness:** 2
**Presentation:** 2
**Contribution:** 2
**Rating:** 2
**Confidence:** 3

**Summary:**

This paper proposes SHARP, a new knowledge distillation framework for compressing large language models. The key idea is to address the problem of "gradient interference" that can occur when distilling knowledge from multiple granularities of the teacher's attention mechanism. The method decomposes attention patterns into token-level, head-level, and layer-level representations. It then projects each of these into a separate, orthogonal rank space to enable independent optimization without conflicting gradients. The authors show through extensive experiments that their method outperforms existing distillation techniques on various NLG and NLU tasks.

**Strengths:**

- The authors provide a comprehensive empirical evaluation of their method. The experiments are conducted on a variety of model architectures, including OPT, LLaMA, and DeepSeek, and are tested across a wide range of benchmarks for both natural language generation and understanding. This thoroughness helps to validate the general applicability of the proposed approach.

**Weaknesses:**

- The core motivation of the proposed method is difficult to understand fully. The paper posits that a high cosine similarity (e.g., 0.68) between different attention representations causes detrimental gradient interference. This is counterintuitive, as a high similarity might be expected to lead to gradients that are similarly aligned rather than conflicting.

- The orthogonality constraint introduced by the method may not be fully effective. The constraint ensures that the gradients are decoupled with respect to the projection matrices. However, it is not clear if this guarantee holds when the gradients are propagated back to the main parameters of the student model.

- The main figure illustrating the SHARP framework is not easy to follow. The diagram is quite dense, and the flow of the distillation process is not immediately clear, making it challenging to understand the mechanics of the approach.

- The choice of the OPT model family for the main results is somewhat outdated. While the paper includes other models, a focus on more recent and widely used architectures for the primary analysis would strengthen the relevance of the findings.

**Questions:**

1.  Have you considered a simpler baseline where the student model is trained to directly mimic the fine-grained attention scores from every head in every layer of the teacher? It seems that if the student can achieve high similarity at this fine-grained level, it would naturally capture the global token dependencies, head-specific patterns, and layer-wise progression that the proposed decomposition aims to separate.

2.  Could you please clarify the memory usage reported in Table 4? The peak memory is listed as around 950MB, but a 0.5B parameter model would require at least 1GB of memory for its weights alone in fp16/bf16.

---

### Note · Authors · 2025-11-14

I have read and agree with the venue's withdrawal policy on behalf of myself and my co-authors.